# Effects of Repeated Cisplatin and Monosodium Glutamate on Visceral Sensitivity in Rats

**DOI:** 10.3390/cells14010026

**Published:** 2024-12-30

**Authors:** Yolanda López-Tofiño, Laura López-Gómez, Marta Martín-Ruíz, Jose Antonio Uranga, Kulmira Nurgali, Gema Vera, Raquel Abalo

**Affiliations:** 1Department of Basic Health Sciences, University Rey Juan Carlos (URJC), 28922 Alcorcón, Spain; yolanda.lopez@urjc.es (Y.L.-T.); laura.lopez.gomez@urjc.es (L.L.-G.); martamruiz5@gmail.com (M.M.-R.); jose.uranga@urjc.es (J.A.U.); 2High Performance Research Group in Physiopathology and Pharmacology of the Digestive System (NeuGut), University Rey Juan Carlos (URJC), 28922 Alcorcón, Spain; 3Working Group of Basic Sciences on Cannabinoids of the Spanish Pain Society, 28046 Madrid, Spain; 4Institute for Health and Sport, Victoria University, Melbourne, VIC 3021, Australia; kulmira.nurgali@vu.edu.au; 5Department of Medicine Western Health, The University of Melbourne, Melbourne, VIC 3010, Australia; 6Regenerative Medicine and Stem Cell Program, Australian Institute for Musculoskeletal Science (AIMSS), Melbourne, VIC 3021, Australia; 7Associated R+D+i Unit to the Institute of Medicinal Chemistry (IQM), Scientific Research Superior Council (CSIC), 28006 Madrid, Spain; 8Working Group of Basic Sciences on Pain and Analgesia of the Spanish Pain Society, 28046 Madrid, Spain

**Keywords:** cisplatin, chemotherapy, monosodium glutamate, visceral pain, colon immunocytes

## Abstract

Cisplatin, a chemotherapeutic drug, is known for causing gastrointestinal disorders and neuropathic pain, but its impact on visceral sensitivity is unclear. Monosodium glutamate (MSG) has been shown to improve gastrointestinal dysmotility and neuropathic pain induced by cisplatin in rats. This study aimed to determine if repeated cisplatin treatment alters visceral sensitivity and whether dietary MSG can prevent these changes. Male Wistar HAN rats were treated with saline or cisplatin (2 mg/kg/week, ip) for 5 weeks, and visceral sensitivity to intracolonic mechanical stimulation was recorded after the final cisplatin administration (week 5) and one-week post-treatment (week 6). In a second cohort, rats treated with cisplatin or saline also received MSG (4 g/L) in their drinking water, and visceral sensitivity was evaluated on week 6. Finally, the untouched distal colon was obtained from a third cohort of animals one week after treatment to assess immunocyte infiltration. Cisplatin significantly increased colonic mechanical sensitivity on week 6 but not on week 5. MSG did not prevent cisplatin-induced visceral hypersensitivity on week 6 and even exacerbated it. On week 6, compared with the control, cisplatin (with or without MSG) did not modify the colonic infiltration of eosinophils, macrophages, neutrophils, or mast cells. Although MSG seems to be useful in ameliorating some of the adverse effects of cisplatin, such as gastrointestinal motility disturbances or neuropathic pain, it does not alleviate visceral pain.

## 1. Introduction

Chemotherapy is one of the most frequently used treatments of cancer but, unfortunately, it is associated with numerous adverse effects. Among others, the most frequent side effects are those affecting the gastrointestinal system (diarrhea, nausea, vomiting, mucositis), anorexia, and peripheral neuropathy, which may affect both the somatic and enteric nervous systems [1,2,3,4,5].

Cisplatin is an antineoplastic drug commonly used in the treatment of different types of cancer and the most emetogenic antitumoral agent. Moreover, it is well known that this drug causes gastrointestinal disorders, such as delayed gastric emptying and gastric distension, associated with satiety sensation and decreased appetite [4,6,7,8,9].

Another symptom typically encountered in cancer patients is pain. Nearly 59% of oncological pain arises from complications associated with cancer treatment [10], including the use of chemotherapeutic agents. In line, in preclinical models, cisplatin given repeatedly (in cycles, like in humans) is well known to cause peripheral neuropathic pain [11,12,13]. In contrast, visceral pain related to chemotherapy has received less attention in both humans and experimental animals. Visceral pain is defined as pain arising from hollow visceral organs, evoked by excessive contraction, stretching, tension, or ischemia of the internal walls [14]. This type of pain is diffuse, difficult to localize, and irradiated to superficial structures [15,16]. It may be associated with mucositis [2,4] and enteric neuropathy [5,17,18] induced by chemotherapy. In addition, in some pathologies, such as irritable bowel syndrome (IBS), it has been shown that variations in the number of immunocytes are associated with visceral sensitivity alterations [19,20]. In fact, some studies suggest that macrophages, mast cells, and neutrophils infiltrate the colon of IBS patients and may favor the production of brain-derived neurotrophic factor in this organ, thus stimulating nerve endings and favoring the appearance of visceral pain [21]. Interestingly, in rodents, a single dose of the antineoplastic drug paclitaxel has been shown to induce acute visceral pain [22]. In contrast, a single high dose of cisplatin decreased the response to intracolonic mechanical stimuli [23]. However, the effects of repeated treatment with cisplatin (comparable to the standard cyclic treatment employed in clinical practice) on visceral pain and the number of colonic immunocytes have not yet been evaluated.

Monosodium glutamate (MSG) is the sodium salt of L-glutamic acid (or glutamate); it is naturally present in a variety of foods [24,25] and is used by the food industry as a flavor enhancer for its characteristic “umami” flavor [26]. Glutamate participates in neurotransmitter synthesis, immunomodulation, and excitatory neurotransmission in the glutamate/GABA-glutamine cycle [27,28,29]. Furthermore, MSG is known to act as an agonist of type 1 gustatory receptors (T1Rs), activating the T1R1/T1R3 heterodimer. Accordingly, some authors have shown that dietary supplementation with MSG suppresses chemotherapy-induced dysgeusia by attenuating the decrease in T1R3 in the tongue of cancer patients [30]. On the other hand, T1R1 and T1R3 subunits have been identified in enteroendocrine cells of the intestine and in the smooth muscle of the stomach and have been shown to participate in colonic motility (accelerating the expulsion of feces in vitro) and in the relaxation of the stomach [31,32]. Interestingly, some authors demonstrated through cine MRI that MSG accelerates gastric emptying and increases duodenal motility in healthy volunteers [33]. Moreover, MSG’s effect has been evaluated in animals repeatedly treated with chemotherapy, specifically with cisplatin, and it has been found to improve some of the parameters altered by this treatment. In particular, it reduced the weight loss of the animals, improved gastrointestinal transit, and prevented the development of peripheral neuropathy [34,35,36], but to our knowledge, its effects on visceral sensitivity associated with chemotherapy have not been described so far.

In view of this background, we aimed to evaluate the effects of a repeated (cyclic) treatment with the antitumoral drug cisplatin on visceral sensitivity (aim 1) and determine if MSG may prevent those effects, as it does with neuropathic pain (aim 2). Finally, immunocyte infiltration in the distal colon was assessed to determine if variations in these cells may contribute to the functional effects (aim 3).

## 2. Materials and Methods

The experiments were carried out at Universidad Rey Juan Carlos (URJC; Madrid, Spain) and were designed and performed according to the EU Directive for the Protection of Animals Used for Scientific Purposes (2010/63/EU) and Spanish regulations (Law 32/2007, RD 53/2013, and order ECC/566/2015) and approved by the Animal Ethics Committee at URJC and Comunidad Autónoma de Madrid (PROEX 061/18). The health and welfare of the animals used for the study was supervised by the personnel of the URJC Veterinary Unit where the study was performed. As humanitarian endpoint criteria, the severity sheets available to researchers in the animal facility were used to assess animal suffering (for example, if the animals’ body weight loss exceeded 10%, the animals would be euthanized). Every effort was made to minimize animal pain and discomfort, as well as to reduce the number of animals utilized in the study.

### 2.1. Animals

Eighty-five young adult (2–3 months) male Wistar HAN healthy rats were used. Rats (215–339 g, n = 8–11/group) were obtained from the Veterinary Unit of URJC. After simple randomization, animals were housed in groups (3–4/cage) in standard transparent cages under environmentally controlled standard conditions, with a 12 h light/12 h dark cycle (lights on at 8.00 a.m.). Confounding factors such as the order of treatments and measurements or animal/cage location were not controlled. Unless otherwise stated, animals had free access to standard laboratory rat chow (Harlan Laboratories Inc., Barcelona, Spain) and sterile tap water.

### 2.2. Experimental Protocol

On the first day of weeks 1–5, the different groups of rats received one intraperitoneal (ip) injection of cisplatin (2 mg/kg, 2.5 mL/kg) or saline (0.9% NaCl *w*/*v*, 2.5 mL/kg). This dose and route of delivery are commonly used in rats to induce a wide range of toxic effects [12,18,36] that are observed in humans and lie within the limits of tolerable toxicity. To reduce cisplatin-induced nephrotoxicity, 2 mL of saline was injected subcutaneously 20 min before ip saline or cisplatin [36]. Body weight was monitored throughout the study.

Three cohorts of animals were used in three different studies (Figure 1). Study 1 was designed to determine the effects of repeated cisplatin on visceral sensitivity (aim 1). Study 2 was designed to evaluate whether MSG (4 g/L) may modify the effects induced by repeated cisplatin on visceral sensitivity (aim 2). Study 3 assessed the number of colonic immunocytes in colon samples using immunohistochemical techniques (aim 3).

The evaluation of the records and histological sections was blinded and “offline”, away from the time of the experiment.

#### 2.2.1. Study 1

Visceral sensitivity was evaluated, as described below, at two different time points in separate groups of animals, namely 2 h after the last cisplatin administration (week 5, cohort 1) or one week after treatment finalization (week 6, cohort 2). This was conducted to determine if the effects associated with the repeated administration of this antitumoral drug in the long term, likely influenced by its neurotoxic effects, could be modified by the effects that occur early after its administration, as seen in our previous study with a single high dose of the compound [23].

Thus, the experimental groups used in this study were Saline WK5 (week 5, n = 11); Cisplatin WK5 (n = 8); Saline WK6 (week 6, n = 8); and Cisplatin WK6 (n = 10).

#### 2.2.2. Study 2

One week after treatment cessation, visceral sensitivity was assessed in rats treated with cisplatin or saline that were exposed or not to MSG (4 g/L) in drinking water from week 0 (one week before the first saline or cisplatin injection) till week 6. This dose of MSG, corresponding to approximately 0.45 g/kg/day (which in turn corresponds to 5.1 g per day for a 70 kg man [37], was previously shown to prevent the development of cisplatin-induced neuropathic pain in the rat, without eliciting significant toxic effects when administered alone [34,36,37].

To reduce the number of animals employed in Study 2, the groups not exposed to MSG (receiving drinking water (W)) were those belonging to cohort 2 in Study 1. Thus, in addition to Saline WK6 (now termed S+W) and Cisplatin WK6 (now termed C2+W) groups, as previously mentioned, we used two more groups (also belonging to cohort 2), which were S+MSG (n = 10) and C2+MSG (n = 8).

#### 2.2.3. Study 3

To reduce the total number of animals employed in this research, in Study 3, colon samples were those collected from animals used in a previously published study [36]. The administration protocol was the same as in Study 2, and samples were collected one week after the cessation of treatment. Animals from Study 2 were not used for this study to avoid using colonic samples potentially damaged or altered by the visceral pain experiments. The samples were used for the analysis of the gut wall and immunocytes (see below).

Thus, the experimental groups (cohort 3) used in this study were as in Study 2, with S+W (n = 6–7), S+MSG (n = 7–8), C2+W (n = 8–9), and C2+MSG (n = 5–7).

### 2.3. Assessment of Colorectal Sensitivity

Colorectal sensitivity was measured once, at the URJC animal facility, as previously described [23]. Briefly, after sedation with Sedator^®^ (medetomidine hydrochloride, 1 mg/kg, ip), a 10 cm longitudinal line was drawn over the linea alba of the abdomen. Transverse lines were drawn every 2 cm to better visualize the abdominal contractions during the recordings. Then, fecal material was gently removed from the rectum and a 5 cm long latex balloon lubricated with Vaseline was inserted through the anus into the colon so that the tip of the balloon was 7 cm inside the colorectum. The catheter to which the balloon was connected was fixed to the tail of the rat with Parafilm^®^ to avoid its expulsion. Sedation was reverted with Revertor^®^ (atipamezole hydrochloride, 0.66 mg/kg, ip). After waking up (normally in <5 min), the abdominal contractions were recorded using a video camera (iPad; Apple, Madrid, Spain) located 30 cm below the recording cage floor. The first 5 min were only used to confirm the normal behavior of the rat after recovery from sedation and was discarded; thereafter, the pressure of the intracolonic balloon was gradually increased using a sphygmomanometer. Tonic stimulation was applied; pressure was increased from 0 to 75 mmHg in steps of 15 mmHg every 5 min and finally returned to 0 mmHg again (for each pressure value, a single stimulus was applied and maintained for 5 min). For each pressure interval, the mean number and duration of contractions was assessed, together with the proportion of time during which the rat’s abdomen contracted.

### 2.4. Evaluation of Colonic Immunocytes

For the analysis of immunocytes, following euthanasia under anesthesia with pentobarbital, performed at the URJC animal facility, 2 cm long samples were obtained from the distal colon, fixed in buffered 10% formalin and embedded in paraffin.

Immunohistochemistry was used to evaluate macrophages and neutrophils. This analysis was performed on colonic paraffin-embedded sections of 5 μm thickness. Deparaffined slides were washed with phosphate-buffered saline (PBS) with 0.05% Tween 20 (Calbiochem, Darmstadt, Germany). Thereafter, sections were incubated for 10 min in 3% (*v*/*v*) hydrogen peroxide to inhibit endogenous peroxidase activity and blocked with horse serum for 30 min to minimize nonspecific binding of the primary antibody. Then, samples were incubated overnight with the following antibodies: mouse anti-rat CD163 (BioRAD; Alcobendas, Spain; MCA342GA, 1/1000), to quantify M2 macrophages and rabbit anti-rat MPO (Abcam; Amsterdam, The Netherlands; ab65871, 1/1000), to quantify neutrophils. The ImmPRESS^®^ HRP (horseradish peroxidase) Universal kit (Vector Laboratories Inc., Burlingame, CA, USA; horse anti-rabbit IgG plus polymer kit or horse anti-mouse IgG PLUS polymer kit) served as the secondary polymer. Samples were counterstained with hematoxylin and coverslip mounted with Eukitt mounting media (O. Kindler GmbH & Co., Freiburg, Germany). The quantification of macrophage and neutrophil numbers was performed all along the submucosa with Fiji-Image J 1.55f National Institutes of Health (NIH) software from a minimum of five micrographs per sample under a 20× objective studied under a Zeiss Axioskop 2 microscope (Zeiss, Tres Cantos, Madrid, Spain).

To evaluate mast cells, colonic sections were stained with toluidine blue [38]. Finally, to assess the presence of eosinophils and general appearance of the colonic wall, sections were stained with hematoxylin–eosin. The number of mast cells and eosinophils was counted under a 20× objective all along the mucosa and submucosa.

### 2.5. Compounds and Drugs

Cisplatin was purchased from Merck Life Science (Darmstadt, Germany) and dissolved in saline (sonicated for about 15 min). Saline or cisplatin volumes were adjusted to a maximum of 2.5 mL/kg. Monosodium glutamate was purchased from Productos Químicos Manuel Riesgo SA (Madrid, Spain) and diluted in filtered sterile water (4 g/L).

### 2.6. Statistical Analysis

Visceral pain is the outcome measure that was used to determine the sample size. The sample size for each experiment was estimated using G*power (release 3.1.9.7) assuming α = 0.05 and power = 0.8 and 2-tailed tests. All the data obtained during the in vivo experiments were included in the statistical analysis, and no animal was excluded from the analysis. In the case of immunocyte analysis, outliers (those deviating by 20% of the standard deviation) were excluded: in the macrophage analysis, one sample from the S+W group was eliminated; in the analysis of neutrophils, two samples were eliminated for being outliers, one from the S+MSG group and another from C2+W; and finally, in the analysis of mast cells, one sample was eliminated for being an outlier in the S+MSG group. Each animal was considered as an experimental unit when analyzing the differences related to body weight, visceral pain, and colonic immunocytes.

Graphs and statistical analyses were performed using GraphPad Prism v. 8.0.2 (GraphPad Software Inc., La Jolla, CA, USA). Normality was assessed by Shapiro–Wilk’s comparison and data were presented as the mean values ± SEM. To compare the normally distributed data, one-way ANOVA was used, followed by post hoc Tukey’s multiple comparison test. In the case of not normally distributed data, the Kruskal–Wallis test followed by Dunn’s multiple comparison test was performed. Values of *p* < 0.05 were considered significantly different.

## 3. Results

### 3.1. Effects of Cisplatin on Colorectal Sensitivity (Study 1)

All animals gained weight progressively throughout the study. The Saline WK5 group tended to gain more weight than the Saline WK6 group (without statistically significant differences), maybe because these groups had a significantly different initial body weight (WK5: 236 ± 2.91 g and WK6: 301 ± 3.80 g, *p* < 0.05, respectively). Both cisplatin-treated groups gained less weight compared to their corresponding control groups, and the differences reached statistical significance from week 3 (Figure 2a). No mortality was recorded.

In the control groups (Saline WK5, Saline WK6), the number of abdominal contractions per minute progressively increased with intracolonic pressure, with a maximum of nine and eight contractions per minute at 75 mmHg, respectively, but the graphs practically overlapped, and the differences were not statistically significant (Figure 2b). Compared to the control animals, the Cisplatin WK5 group tended to have more abdominal contractions in response to the lower pressures, although the values were similar to those in the control groups from pressure 45 mmHg onwards; the differences with the Saline WK5 group did not reach statistical significance at any pressure. In contrast, the Cisplatin WK6 group presented a higher number of contractions than its control (Saline WK6) during the whole study, the differences being statistically significant at 30 and 60 mmHg. At 60 mmHg, it reached a maximum of 10 contractions per minute, whereas the control groups showed around seven contractions per minute.

The duration of the contractions (Figure 2c) was around 2.5–3.0 s at all pressures in all experimental groups. Therefore, the percentage of time that the rats displayed abdominal contractions (time in contraction, Figure 2d), showed a very similar profile to the number of contractions. Thus, control groups presented a progressively longer time in contraction correlated with the level of intracolonic pressure, with a maximum of 33% at 75 mmHg. The group Cisplatin WK5 tended to increase the time in contraction with respect to the control groups at low pressures, but these values overlapped with those of the control groups at high pressures (45–75 mmHg). Cisplatin WK6 tended to increase the time in contraction with respect to the control and Cisplatin WK5 groups, although the differences did not reach statistical significance.

Both cisplatin groups tended to increase the number of contractions and the time in contraction at the initial pressure 0 (in the absence of intracolonic mechanical stimulation) but without statistically significant differences with any of the other groups (Figure 2b–d). In all groups, the number of contractions and time in contraction decreased to basal values when intracolonic pressure was reduced back to 0 mmHg, without any statistically significant difference among them.

### 3.2. Effects of Monosodium Glutamate on Colorectal Sensitivity (Study 2)

As mentioned above, compared to water, cisplatin tended to produce a decrease in body weight gain, although without the differences reaching statistical significance. MSG alone (S+MSG) tended to increase the body weight of saline-treated animals without statistically significant differences with the control (S+W). The double treatment C2+MSG overlapped with the group treated with cisplatin alone (C2+W), meaning that MSG could not counteract the effect of chronic cisplatin treatment (Figure 3a). As in study 1, no mortality was recorded in any of the experimental groups.

In this study, colorectal sensitivity was evaluated only during WK6 (one week after the end of cisplatin treatment, cohort 2). As described above, compared to controls, cisplatin (C2+W) produced an increase in abdominal twitch responses when visceral sensitivity was assessed one week after the end of treatment. The number of contractions per minute in the S+MSG group (Figure 3b) showed a progressive increase as intracolonic pressure increased, with a maximum of 10 contractions per minute at 75 mmHg. Although values were somehow higher than those of the S+W group (control) at all pressures, the differences did not reach statistical significance. The C2+MSG group presented a higher number of contractions during the whole experiment, with statistically significant differences with the S+W group at 0–45 mmHg (at these pressures, the mean values were also higher than those of C2+W and S+MSG, but the differences with these groups did not reach statistical significance). The groups treated with MSG did not reach basal values when intracolonic pressure was set to 0 mmHg again, showing statistically significant differences with the groups that did not receive MSG in the drinking water.

In accordance with Study 1, the duration of contractions (Figure 3c) was around 2.5–3.0 s at all pressures and in all experimental groups, without statistically significant differences among them. Therefore, again, the graphs corresponding to the parameter “Time in contraction” closely resembled those obtained for the number of contractions per minute. Thus, compared to the control group (S+W), S+MSG tended to increase the percentage of time in contraction at low pressures but without statistically significant differences at any of the pressures tested (Figure 3d). In addition, the double treatment with cisplatin and MSG tended to increase this parameter at low pressures, the differences with the control group reaching statistical significance at 0–15 mmHg. The groups treated with MSG (S+MSG and C2+MSG) did not reach as low of values as those not receiving the food additive (S+W and C2+W) when pressure was set back to 0 mmHg at the end of the experiment and showed statistically significant differences with respect to these groups.

### 3.3. Effects of Cisplatin and Monosodium Glutamate on Colonic Immunocytes (Study 3)

Treatment with cisplatin caused some erosion of the apical area of the colonic glands, which was improved when MSG was added to the cisplatin-treated animals. No damage was seen in the MSG-only group (Appendix A). The presence of eosinophils was very low in all cases and did not differ between experimental groups.

M2 macrophages immunoreactive to CD-163 tended to increase in the distal colon of animals exposed to MSG in their drinking water, in the presence or not of cisplatin, without statistically significant differences with the control (Figure 4a).

In contrast, the number of MPO-positive neutrophils tended to decrease in the cisplatin-treated group, and co-treatment with MSG tended to normalize these values compared to the control group, although the differences were not statistically significant (Figure 4b).

Regarding mast cells, the groups treated with MSG or with cisplatin tended to display a decreased number of these cells, although only the group S+MSG presented statistically significant differences with the S+W group, whereas the group that received the double treatment presented values similar to those of the control group (S+W) (Figure 4c).

## 4. Discussion

In this study, we examined the impact of repeated cisplatin administration on rat visceral sensitivity at two time points, shortly after (two hours) and one week following the last dose. Our findings suggest that repeated treatment with this antitumoral drug increases visceral sensitivity; however, this effect may be countered by its immediate inhibitory effects on visceral pain pathways. Interestingly, despite MSG’s known preventive effect on somatic neuropathic pain and gastrointestinal dysmotility induced by this antitumoral drug, this food additive did not prevent but even increased cisplatin-induced visceral hypersensitivity.

### 4.1. Effects of Cisplatin on Body Weight Gain and Visceral Pain

As a positive control in our study, we assessed the changes in body weight gain. Consistent with prior findings following both acute and repeated administration with cisplatin [5,7,23,36], chronic treatment with this antineoplastic drug led to reduced body weight gain, and it likely caused decreased food intake, gastrointestinal dysmotility, and peripheral and enteric neuropathy, as observed in our previous studies [5,7,12].

Other effects of cisplatin, such as alterations in visceral sensitivity, have not been deeply studied thus far. Almost 59% of cancer patients report that they have experienced visceral pain induced by chemotherapy or metastasis [39]. These findings indicate a significant prevalence of visceral pain among cancer patients and underscore the need for improved strategies to enhance their quality of life. In Study 1, control rats (Saline WK5 and Saline WK6) showed an increase in the responses to intracolonic mechanical stimulation (number of contractions and time in contraction) as intracolonic pressure increased, irrespective of the time point at which the experiment was performed, which is consistent with our previously published data [23]. In addition, the groups treated with cisplatin tended to increase these parameters already in the absence of intracolonic mechanical stimulation (pressure 0) at the beginning of the experiment, suggesting the presence of some degree of visceral mechanical allodynia. However, the duration of contractions was homogeneous in all groups, suggesting that abdominal muscle motor function itself was not altered by treatment with cisplatin.

Interestingly, one week after treatment finalization, cisplatin increased the responses to mechanical intracolonic stimulation (and the effect was statistically significant for the number of contractions), a finding resembling the effect of cisplatin on somatic pain thresholds, associated with the development of peripheral neuropathy [11,12,36]. These results suggest that similar neurotoxicity processes may underlie cisplatin-induced somatic and visceral hypersensitivity. It has been suggested that platinum derivatives are intensely neurotoxic to the dorsal root ganglia (DRG, related with peripheral pain) and the area postrema (related to nausea and vomiting) [5], which are not protected by the blood–brain barrier and are highly vulnerable to their effects. It has been proposed that the main mechanism of neurotoxicity of these agents affecting the DRGs is the formation of DNA adducts [40,41], which limits proper transcription. This might damage DRG nerve cells and/or increase the oxidative stress in the mitochondria, with consequent damage to axonal transport [5]. In addition, platinum derivatives have been associated with neuroinflammation [5,42].

Of relevance for the present study, in addition to peripheral neuropathy affecting DRG neurons, cisplatin also produces enteric neuropathy [17]. Furthermore, in anesthetized animals subjected to intracolonic mechanical stimulation, Vera et al. [17] demonstrated that a lower dose of cisplatin (1 mg/kg) produced an increase in colonic contractions at low pressures, whereas a higher dose (3 mg/kg), which had a big impact on the general health of the animals, also decreased their capability to produce colonic contractions one week after treatment cessation. Cisplatin’s impact on colonic [17] and abdominal contractions (present study) could be interconnected, with colonic activity aimed at expelling the intracolonic balloon, potentially triggering colicky pain and increasing abdominal contractions. Additionally, in rats, we observed elevated mRNA of nNOS in the small intestine [18] and an increase in myenteric neurons immunoreactive to nNOS in the colon [17,36], along with a decrease in those immunoreactive to CGRP (calcitonin gene-related peptide) [17] one week after cisplatin treatment cessation (WK6). While other factors akin to peripheral neuropathy, especially those affecting the DRG [5], may also be at play, cisplatin-induced enteric neuropathy [17,18,36] likely contributed to the heightened colorectal sensitivity seen here (WK6).

Alternatively, this increased sensitivity caused by cisplatin may be due to alterations in colonic immune cell infiltration, as occurring in IBS patients. In particular, macrophages, mast cells, and neutrophils have been shown to produce inflammatory cascades associated with disturbances in the gut–brain interaction [14,43,44]. Mast cells release mediators such as histamine and cytokines, which can increase smooth muscle contraction and nociceptor sensitization, increasing pain perception and visceral sensitivity [45]. Similarly, neutrophils release cytokines and chemokines that recruit and activate other immune cells, contributing to tissue damage and inflammation [46]. Finally, there are macrophages with pro-inflammatory and pronociceptive (M1) and anti-inflammatory and antinociceptive (M2) effects [47], and their imbalance in the colon could increase visceral sensitivity. Regarding eosinophils, their presence in the colonic mucosa has been described in cases of infectious enterocolitis, medication-related injuries, radiation effects and inflammatory bowel disease [48]. In this work, no evidence of significant alterations in the number of any of the immunocytes evaluated was found (probably due to the scarce mucosal damage observed), discarding a major contribution of them to the increased visceral sensitivity caused by cisplatin.

In contrast to the effects observed at WK6, evaluating colorectal sensitivity just 2 h after the last cisplatin administration (WK5) revealed only minor and non-statistically significant differences compared to control animals. Notably, cisplatin seemed to enhance contractions at lower pressures, but by 45 mmHg, the values aligned with the control group. It is unlikely that neurotoxic effects were absent in this group, as they had already received four prior injections plus a fifth one. Rather, immediate inhibitory effects following cisplatin administration likely masked the neurotoxicity’s impact. Interestingly, our previous research indicated a reduction in abdominal contractions in response to intracolonic mechanical stimulation just 2 h after a single high dose of cisplatin in otherwise healthy animals (not suffering from any neurotoxicity) [23], suggesting that the immediate effect of cisplatin is not an increase (as shown for paclitaxel [22]) but a reduction in visceral sensitivity. This effect may explain the lack of increased visceral sensitivity in animals whose colons, most likely, were already sensitized by the four previous cisplatin administrations, associated with cisplatin-induced peripheral (including enteric) neurotoxicity, as discussed above.

One significant adverse effect of cisplatin is nausea and vomiting [49,50,51]. In rats, which cannot vomit, other related markers are patent, such as gastric distension and pica (ingestion of non-nutritive substances in response to emetogenic stimuli, like cisplatin) [7,23,52]. These effects result from serotonin release (and serotonin-based antiemetics effectively counteract vomiting in humans [53]) and gastric distension and pica in rats [23]), which activates the vagus nerve [53]. Notably, some authors [54,55] suggested that acute luminal serotonin may decrease visceral nociception, potentially explaining hyposensitivity in our previous study [23] and the lack of hypersensitivity in this study, possibly due to an immediate inhibitory effect of the fifth cisplatin administration. Intriguingly, chronic luminal serotonin has the opposite effect, causing hyperalgesia, which may contribute to the results observed during WK6, one week after treatment finalization [54,55].

Interestingly, some researchers suggest that acute serotonin-induced hyposensitivity could be due to serotonin-induced anandamide (AEA) release and the subsequent activation of cannabinoid receptor 1 (CB_1_R) [55]. However, Feng et al. [55] highlighted that chronic luminal serotonin-induced hypersensitivity is linked to decreased AEA levels, suggesting that changes in the endocannabinoid system activity may play a role in the reduction in visceral pain shortly after cisplatin administration and its increase over time. On the other hand, 5-HT_7_ receptor antagonists have been shown to alleviate intestinal hyperalgesia and reduce neurotrophin formation in IBS [56]. Additionally, research by Ben Rehouma et al. [57] indicated that the vagus nerve blockade (vagotomy) inhibits acute pain but results in increased microglial cells 1–2 days after insult, which is associated with the release of pro-inflammatory cytokines that contribute to visceral pain transmission. Interestingly, electroceuticals that activate the vagus nerve can decrease visceral sensitivity [57,58,59].

Altogether, diverse factors may contribute to visceral pain perception in cisplatin-treated animals; therefore, the immediate inhibitory effects of cisplatin on visceral sensitivity may result from vagus nerve activation via serotonin release, alterations in endocannabinoid synthesis and release, or changes in microglia activity. However, it is possible that other mechanisms, such as the activation of descending pain inhibitory pathways involving opioids [60] may also play a role in these effects. Further research is warranted to elucidate the precise mechanism of action involved in our results.

### 4.2. Effects of MSG on Cisplatin-Induced Alterations

In Study 2, MSG did not influence body weight gain and could not reverse the tendency of weight loss induced by cisplatin, in contrast with previous reports [36,37], suggesting that the effect of MSG on this parameter (at least when MSG is included in the drinking water) is not robust but particularly sensitive to the specific experimental conditions.

Our earlier work demonstrated MSG’s effectiveness in improving gastrointestinal dysmotility and alleviating neuropathic pain caused by cisplatin [36]. In addition, when MSG was administered to vincristine-treated animals, responses to intracolonic mechanical stimulation were reduced [61], motivating us to explore its effects on cisplatin-induced visceral sensitivity changes.

Due to the heightened visceral pain observed one-week post-cisplatin treatment, we assessed MSG’s impact at this time point (WK6) to determine if it could prevent the development of visceral hypersensitivity.

In this study, MSG alone tended to increase the visceral pain responses. This seems to be in accordance with López-Miranda et al. [37], who found that MSG raised colonic contractions, likely through its direct interaction with umami taste receptors in the colonic mucosa, potentially causing colicky pain. Importantly, glutamate is the main excitatory neurotransmitter of the nervous system and acts on metabotropic (mGluRs) and ionotropic receptors [62]. Remarkably, glutamatergic signaling has been shown to be important in pain transmission and sensation [63]. This might contribute to explaining our findings.

Despite its neuroprotective effect against somatic neuropathic pain [34,35,36], MSG did not prevent the increase in the abdominal contractions in response to mechanical intracolonic stimulation in cisplatin-treated animals. Furthermore, it increased those produced at low pressures with statistically significant differences compared to the control group. The above-mentioned effects of MSG on colonic motor function [37] may have contributed to these findings. However, other mechanisms, more related to modulation of pain pathways, may have also participated. For example, mGluR5 has been found in the gastroesophageal vagal pathways involved in the peripheral excitatory modulation of vagal afferent mechanosensitive input [64,65,66]. In addition, mGluR1 and mGluR5 have been found in the submucosal plexus [67,68]. Remarkably, some authors suggest that the decreased expression of mGluR5 in enteric glia is related to inflammation [69] and that this receptor participates in the mediation of visceral nociception, favoring this type of pain when this receptor is activated [70]. Therefore, the addition of MSG in the diet may facilitate the activation of these receptors in the gastrointestinal tract and thus the increase in visceral sensitivity. It is important to mention that MSG is considered a toxin when used at high doses [71], although at doses accepted for consumption, it does not cause relevant health problems [37,71] due to its low oral bioavailability, exerting its effects mainly locally.

Another possible mechanism for visceral hypersensitivity caused by MSG could be related to inflammation [71]. In our previous study, we observed some damage of the small intestinal wall occurring one week after cisplatin treatment finalization (not related with mucosal turnover, evaluated through Ki-67 immunohistochemistry, which was not increased [18]), and this was not improved by MSG, which did not cause any histological damage per se [36]. In the present study, similar damage was seen in the apical area of the colonic glands after cisplatin treatment, although in this case, MSG administration seemed to have a protective effect. However, MSG tended to increase the number of macrophages in the colon (irrespective of cisplatin being present or not), although MSG alone did not induce morphological alterations. This is consistent with the findings by other authors in neonatal rats [72]. Neutrophils and eosinophils did not change and, curiously, mast cells decreased (in a statistically significant manner) in the colon samples from S+MSG (but not in C2+MSG) group. Changes in immunocyte infiltration, in any case, were only minor. Thus, the low level of intestinal inflammation does not seem to be sufficient to explain the increased visceral sensitivity of animals treated with MSG and cisplatin.

## 5. Conclusions

This study demonstrates that cisplatin induces visceral hypersensitivity to intracolonic mechanical stimulation in the rat, resembling neuropathic pain symptoms (mechanical allodynia in the paws [36]), likely through neurotoxicity mechanisms. However, inhibitory mechanisms activated shortly after its administration appear to mitigate visceral hypersensitivity. These mechanisms may involve serotonin release, vagal responses, endocannabinoid system, and microglia activation, among other possibilities.

On the other hand, while regular MSG intake in drinking water prevents neuropathic pain and also alleviates gastrointestinal dysmotility to some extent [36], it does not have the same effect on cisplatin-induced visceral hypersensitivity. This suggests that MSG’s mechanisms of action differ in somatic and visceral nociception. Local, direct effects of MSG on motor function and pain perception may contribute to its impact on visceral nociceptive responses. Furthermore, the effects of MSG may depend on the particular antitumoral drug causing the symptoms (Figure 5 shows a summary of the effects observed for both cisplatin, here and in our previous research [36], and vincristine elsewhere [61]).

Further preclinical studies are needed to ascertain the mechanisms of action involved in the MSG effects counteracting those of antineoplastic drugs. Whatever the mechanism involved, though, our results highlight the interest of performing specific clinical studies aimed at clarifying the effectiveness of MSG to improve the quality of life of cancer patients suffering from chemotherapy-induced adverse effects.

## Figures and Tables

**Figure 1 cells-14-00026-f001:**
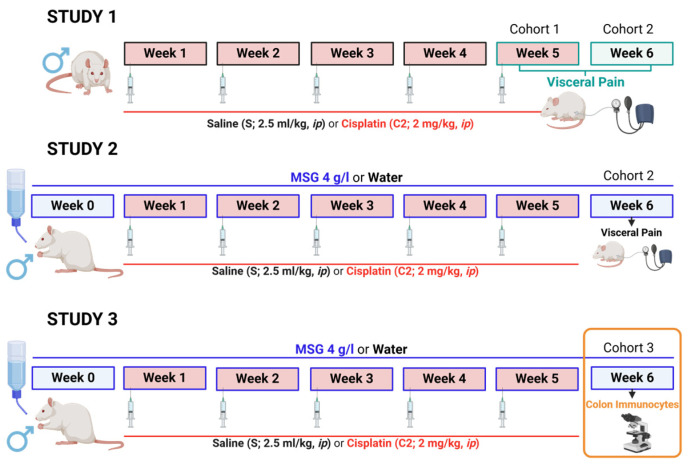
Experimental protocol. Visceral pain was evaluated in two cohorts of rats that were intraperitoneally (ip) administered with saline (2.5 mL/kg) or cisplatin (2 mg/kg, C2) for 5 consecutive weeks (weeks 1–5). Study 1: visceral sensitivity was recorded after the fifth week (week 5) or one week after treatment (week 6) in separate groups of animals (cohorts 1 and 2, respectively). Study 2: animals treated with saline or cisplatin were exposed or not to monosodium glutamate (MSG, 4 g/L) in drinking water from week 0 to 1 week after treatment (cohort 2), and visceral sensitivity was recorded one week after treatment (week 6). Study 3: untouched distal colon samples were collected from a separate group of animals (cohort 3) receiving the same treatments as cohort 2 and evaluated on week 6 using histological and immunohistochemical analyses of mast cells, macrophages, and neutrophils. Created in BioRender. Neugut. (2024) BioRender.com/d69z388, accessed on 6 September 2024.

**Figure 2 cells-14-00026-f002:**
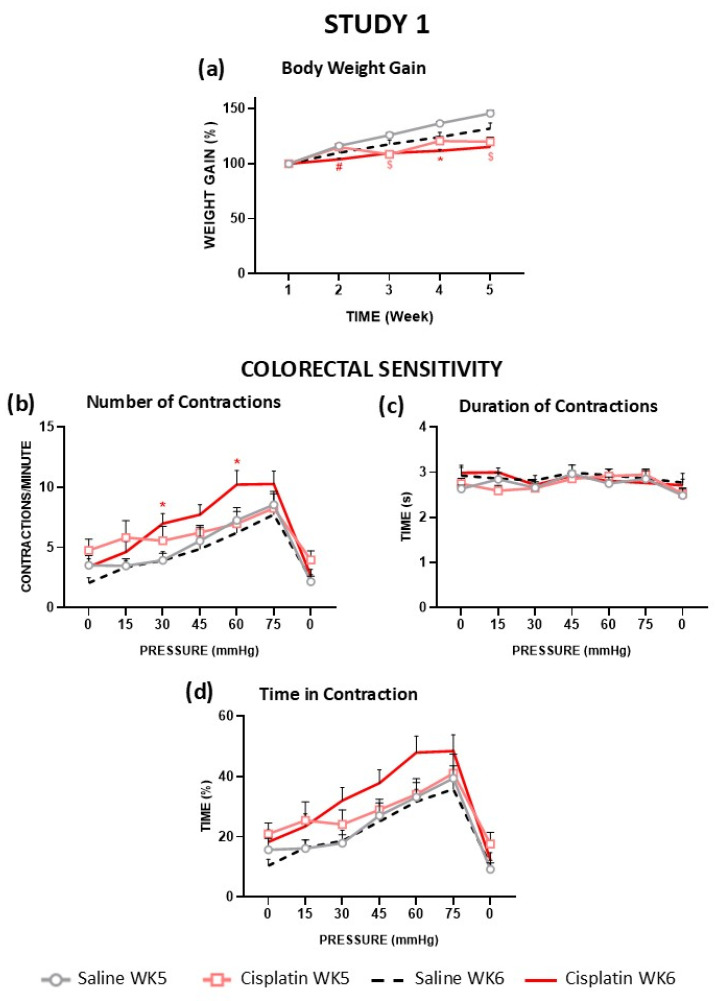
Effect of cisplatin on body weight gain and visceral sensitivity in the rat (Study 1). The rats were intraperitoneally administered with saline (2.5 mL/kg) or cisplatin (2 mg/kg) for 5 consecutive weeks (weeks 1–5). The body weight gain (**a**) of animals was measured along treatment. Visceral sensitivity was recorded 2 h after the fifth injection (week 5, WK5) or one week after it (week 6, WK6) in separate groups of animals (cohorts 1 and 2, respectively). Animals were subjected to tonic mechanical intracolonic stimulation. Pressure was increased from 0 to 75 mmHg in steps of 15 mmHg every 5 min to finally return to 0 mmHg again; for each pressure value, a single stimulus was applied and maintained for 5 min. The number of contractions (**b**), duration of contractions (**c**), and time in contraction (**d**) were measured. Experimental groups were Saline WK5 (gray line, n = 8), Cisplatin WK5 (pink line, n = 8), Saline WK6 (black dashed line, n = 11), or Cisplatin WK6 (red line, n = 10). Data represent the mean ± SEM. $ *p* < 0.05 vs. Saline WK5; * *p* < 0.05 vs. Saline WK6; # *p* < 0.05 vs. Cisplatin WK5 (one-way ANOVA followed by Tukey’s post hoc test or the Kruskal–Wallis test followed by Dunn’s multiple comparison test as appropriate).

**Figure 3 cells-14-00026-f003:**
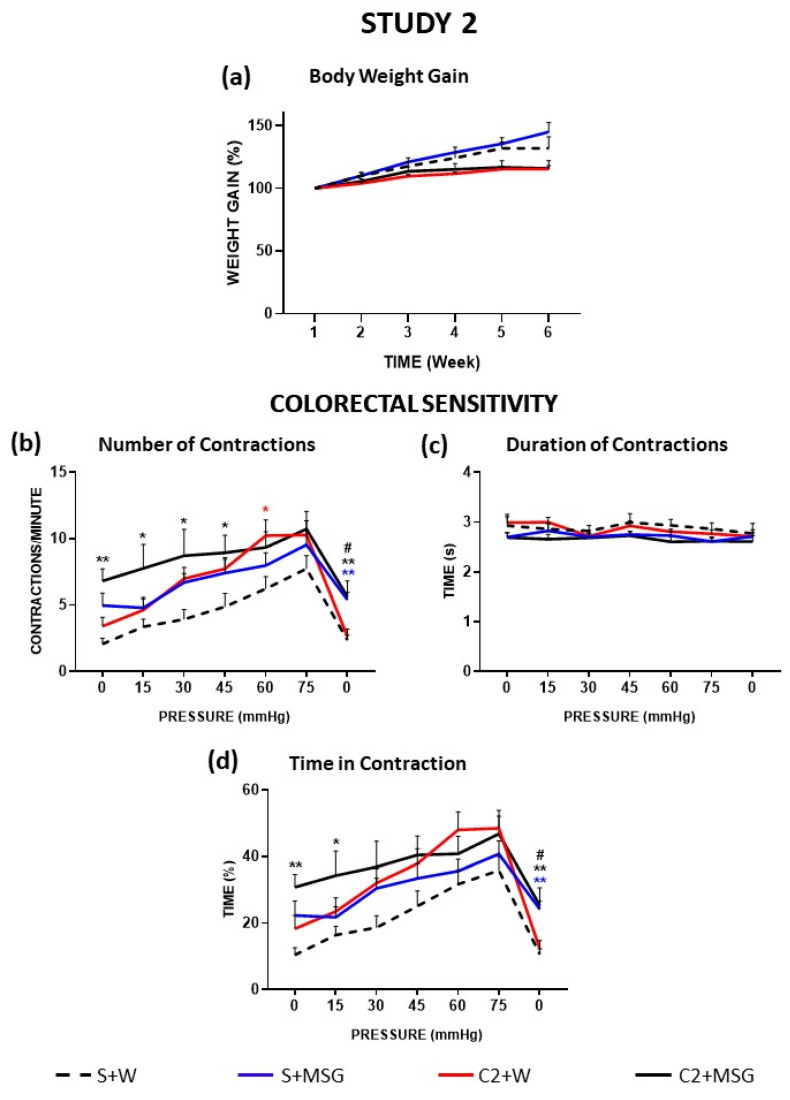
Effect of cisplatin and monosodium glutamate (MSG) on body weight gain and visceral sensitivity in the rat (Study 2). The rats were intraperitoneally administered with saline (2.5 mL/kg, S) or cisplatin (2 mg/kg, C2) for 5 consecutive weeks (weeks 1–5) and exposed or not to MSG (4 g/L) in drinking water (W) from week 0 to 1 week after treatment (week 6). The body weight gain (**a**) of animals was measured along treatment and post-treatment. Visceral sensitivity was recorded one week after treatment (week 6). Animals were subjected to tonic mechanical intracolonic stimulation. Pressure was increased from 0 to 75 mmHg in steps of 15 mmHg every 5 min to finally return to 0 mmHg again; for each pressure value, a single stimulus was applied and maintained for 5 min. The number of contractions (**b**), duration of contractions (**c**), and time in contraction (**d**) were measured. Experimental groups were S+W (black dashed line, n = 11), S+MSG (blue line, n = 10), C2+W (red line, n = 10), or C2+MSG (black line, n = 8). Data represent the mean ± SEM. * *p* < 0.05, ** *p* < 0.01 vs. S+W; # *p* < 0.05 vs. C2+W (one-way ANOVA followed by Tukey’s post hoc test or the Kruskal–Wallis test followed by Dunn’s multiple comparison test as appropriate).

**Figure 4 cells-14-00026-f004:**
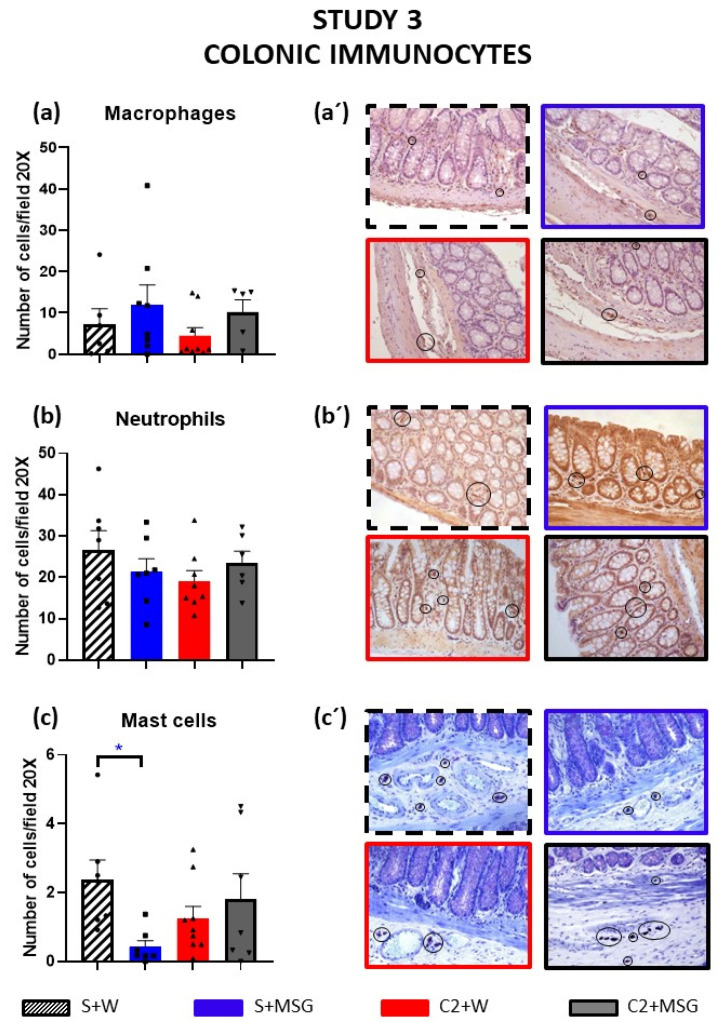
Effect of cisplatin and monosodium glutamate (MSG) on the infiltration of immune cells in the rat distal colon (study 3). The rats were intraperitoneally administered with saline (2.5 mL/kg, S) or cisplatin (2 mg/kg, C2) for 5 consecutive weeks (weeks 1–5) and exposed or not to MSG (4 g/L) in drinking water (W) from week 0 to 1 week after treatment (week 6). At the end of the experiment (week 6), distal colon samples were embedded in paraffin, sectioned, stained with toluidine blue (to study mast cells) or immune-stained with antibodies against CD163 (to study macrophages) or MPO (to study neutrophils), and studied under a Zeiss Axioskop 2 microscope. The number of macrophages (**a**), neutrophils (**b**), and mast cells (**c**) were counted. Photomicrographs of macrophages (**a’**), neutrophils (**b’**), or mast cells (**c’**) taken under the 20× objective are shown. Experimental groups were S+W (n = 6–7), S+MSG (n = 7–8), C2+W (n = 8–9), or C2+MSG (n = 5–7). Data represent the mean ± SEM. * *p* < 0.05 vs. S+W (one-way ANOVA followed by Tukey’s post hoc test or the Kruskal–Wallis test followed by Dunn’s multiple comparison test as appropriate).

**Figure 5 cells-14-00026-f005:**
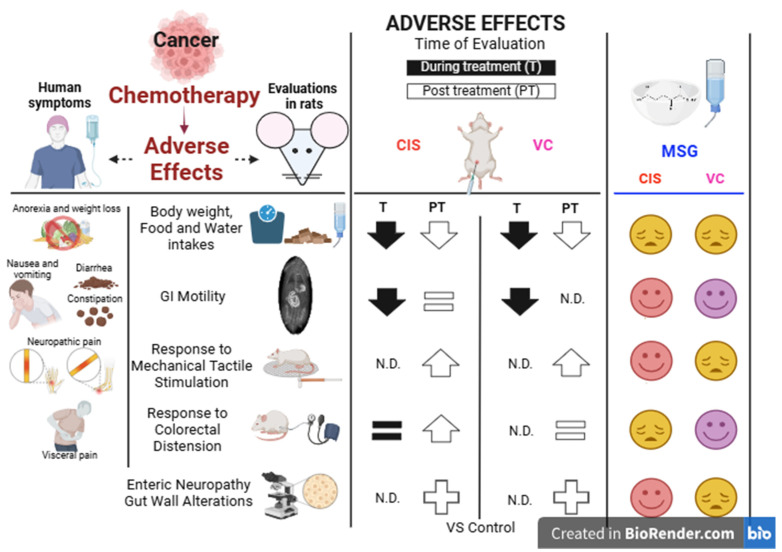
Summary of the adverse effects of the antitumoral drugs cisplatin and vincristine on gastrointestinal motility and sensitivity, and the main results obtained with monosodium glutamate. Our results show that both antitumoral drugs reduced body weight and water and solid intakes during treatment, with an only partial recovery one week after treatment cessation. Cisplatin (CIS) also produced alterations in gastrointestinal (GI) motility during the time of treatment that were maintained one week after termination, whereas vincristine (VC) only produced GI motility alterations during treatment. Both antitumoral agents produced signs of neuropathic pain, measured as the response to mechanical tactile stimulation one week after the end of treatment, but only CIS increased visceral sensitivity to colorectal distension at the same time point. CIS decreased intestinal wall thickness, whereas VC increased it. Monosodium glutamate (MSG) could not improve the alterations produced by both antitumoral drugs on body weight and food and water intakes, but it could ameliorate GI dysmotility. MSG improved the tactile nociceptive alterations induced by CIS but not those induced by VC. On the contrary, MSG was not able to counteract the increased visceral pain induced by CIS but decreased the mechanically stimulated abdominal contractions in animals treated with VC. MSG improved the alterations produced in the thickness of the layers of the GI tract by CIS but not by VC. MSG improved the enteric neuropathy produced by CIS (its effect on the VC-induced enteric neuropathy was not evaluated here). Abbreviations: CIS: cisplatin; GI: gastrointestinal; MSG: monosodium glutamate; VC: vincristine; N.D.: not determined. For more detailed information, see the original articles: CIS [36], VC [61]. Created with Biorender.

## Data Availability

The data presented in this study are available upon request from the corresponding authors and will also be available at the corresponding authors’ institutional repository (URJC).

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
