# Peer review of "Effects of Repeated Cisplatin and Monosodium Glutamate on Visceral Sensitivity in Rats"

_cells, 2024, doi:10.3390/cells14010026_

Round 1
Reviewer 1 Report
Comments and Suggestions for Authors
Thanks a lot for the opportunity to score this ms. This is an interesting and original study dealing with an aspect that has not been investigated so far (cisplatin and visceral pain). Overall the ms is well written, methodology is appropriate and conclusions are well substantiated.
I have few suggestions that may help to improve the ms.
1) Please add info on the physiological action of MSG in the gut.
2) Please add info on the use of MSG on humans if any
3) Discussion is well conceived but partly speculative on the effect of MSG.
4) Whereas the cisplatin dose is justified this is lacking for MSG.
5) Conclusions should be damped on the effects of MSG in ameliorating the effects of chemotherapy unless an adequate translation to humans is possible on the data and discussed
Reviewer 2 Report
Comments and Suggestions for Authors
Manuscript entitled "Effects of repeated cisplatin and monosodium glutamate on visceral sensitivity in rats"
This work is of interest. While some modifications should be made before it can be accepted formally.
Major issues:
1. The quality of figure-4, which is one of the most important figures, is not ideal:
a. The authors should improve the quality of IHC. The current figures show very strong background staining. It is not possible to count the number of cells precisely.
b. The authors should also count the number of eosinophils.
2. Is there any mucosal erosion identified? If yes, the authors should show that.
3. The authors are encouraged to perform TUNEL and/or Ki-67 staining on the colon mucosa to see the status of mucosal turnover.
Comments on the Quality of English LanguageAccepatble
Round 2
Reviewer 1 Report
Comments and Suggestions for Authors
Authors satisfied my requests.
Thank you